# A Novel Screening Strategy Reveals ROS-Generating Antimicrobials That Act Synergistically against the Intracellular Veterinary Pathogen *Rhodococcus equi*

**DOI:** 10.3390/antiox9020114

**Published:** 2020-01-28

**Authors:** Álvaro Mourenza, José A. Gil, Luís M. Mateos, Michal Letek

**Affiliations:** Department of Molecular Biology, Area of Microbiology, University of León, 24071 León, Spain; amouf@unileon.es (Á.M.); jagils@unileon.es (J.A.G.)

**Keywords:** *Rhodococcus equi*, antimicrobials, oxidative stress, macrophages, roGFP2

## Abstract

*Rhodococcus equi* is a facultative intracellular pathogen that causes infections in foals and many other animals such as pigs, cattle, sheep, and goats. Antibiotic resistance is rapidly rising in horse farms, which makes ineffective current antibiotic treatments based on a combination of macrolides and rifampicin. Therefore, new therapeutic strategies are urgently needed to treat *R. equi* infections caused by antimicrobial resistant strains. Here, we employed a *R. equi* mycoredoxin-null mutant strain highly susceptible to oxidative stress to screen for novel ROS-generating antibiotics. Then, we used the well-characterized Mrx1-roGFP2 biosensor to confirm the redox stress generated by the most promising antimicrobial agents identified in our screening. Our results suggest that different combinations of antibacterial compounds that elicit oxidative stress are promising anti-infective strategies against *R. equi*. In particular, the combination of macrolides with ROS-generating antimicrobial compounds such as norfloxacin act synergistically to produce a potent antibacterial effect against *R. equi*. Therefore, our screening approach could be applied to identify novel ROS-inspired therapeutic strategies against intracellular pathogens.

## 1. Introduction

*Rhodococcus equi* is an intracellular pathogen that causes granulomatous infections in a wide range of animal species, including foals, dogs, pigs, cattle, sheep, and goats [1,2,3]. In addition, *R. equi* can infect immunocompromised humans causing a fatal pyogranulomatous bronchopneumonia [4]. *R. equi* is a widely distributed pathogen, and is frequently endemic in horse farms [5,6]. This pathogen is usually transmitted by inhaling aerosol respiratory particles or dust contaminated with *R. equi* [5,6]. 

During the last decade, the number of *R. equi* isolates resistant to commonly used antimicrobials has dramatically increased [7,8,9], making current antibiotherapies often ineffective [10]. Primary preventative strategies based on the use of vaccines or hyperimmune plasma administration do not confer full protection against this pathogen [11,12,13]. Thus, new treatments based on a combination of clinically established antimicrobials with novel anti-*R. equi* compounds could be the only realistic option to solve in the short term the crisis caused by antimicrobial resistant strains. 

Interestingly, some antimicrobial agents can stimulate the production of reactive oxygen species (ROS) as part of their mechanism of action [14,15,16]. Therefore, the antioxidant systems of bacterial pathogens could be important to counteract antibiotic ROS production. We recently demonstrated that mycoredoxins are key factors for the maintenance of *R. equi*’s redox homeostasis [17]. In particular, a *R. equi* mutant strain lacking three mycoredoxin-encoding genes (*R. equi Δmrx1Δmrx2Δmrx3*) shows high susceptibility to oxidative stressing agents, including hydrogen peroxide, sodium hypochlorite, or nitric oxide [17]. Based on this evidence, we used here this triple *mrx*-null mutant strain of *R. equi* to screen for novel ROS-generating antimicrobial compounds. As a result, we identified potent and synergistic antimicrobial activities of different ROS-producing antibiotics against *R. equi*. In addition, we established a novel screening approach for the identification of ROS-inspired antimicrobial strategies against this intracellular veterinary pathogen.

## 2. Materials and Methods

### 2.1. Bacterial Strains, Reagents, and Antibiotic Susceptibility Tests

*Rhodococcus equi* 103S^+^ was kindly provided by Dr. Jesús Navas from the University of Cantabria (Spain), which was considered the wild type strain in all assays. *R. equi Δmrx1Δmrx2Δmrx3* and *R. equi Δmrx1Δmrx2Δmrx3* expressing *mrx1*-*roGFP2* are derivative strains of *R. equi* 103S^+^ [17]. All reagents were procured from Sigma-Aldrich, unless otherwise stated. 

For antibiotic susceptibility tests, exponential growth phase cultures (OD_600_ = 1) of *R. equi* were diluted in melted (at 50 °C) trypticase soy agar (TSA) plates at 1:10 mL dilution and spread over 10 mL solid TSA plates. Disks with antibiotics were then placed onto *R. equi*-containing TSA plates, which were incubated at 30 °C for 24 h.

### 2.2. In vivo Quantification of Redox Homeostasis 

The biosensor Mrx1-roGFP2 [17] was used to evaluate the intracellular Redox potential of *R. equi* 103S^+^ during infection. The redox status of Mrx1-roGFP2 was measured as described previously [17]. Briefly, pellets of exponentially growing *R. equi Δmrx1Δmrx2Δmrx3* expressing *mrx1*-*roGFP2* [17] were resuspended in Phosphate-Buffered Saline (PBS) with different concentrations of antibiotics, treated with 10% N-Ethylmaleimide (NEM) to block free thiol groups, and fixed with 70% ethanol on poly-l-lysine-treated microscope slides. In parallel, the Mrx1-roGFP2 total reduction or oxidation status was calculated by adding 40 mM dithiothreitol (DTT) or 10 mM of diamide (DIA), respectively.

In all cases, ~200 bacterial cells from different treatments were evaluated on a Zeiss LSM800 confocal microscope with Airyscan as previously described [17,18,19]. The ratio between fluorescence emission at 530 nm after excitation at 405 (oxidized) and 490 nm (reduced) was calculated pixel by pixel with ImageJ (http://rsb.info.nih.gov/ij/). The quantification of redox potentials was performed as described by Gutscher et al. [20].

### 2.3. Minimum Inhibitory Concentration (MIC) Assays

The MIC of each antibiotic or combination of antibiotics was calculated as previously described [21]. Briefly, exponentially growing *R. equi* 103S^+^ in a Muller-Hinton broth medium (OD_600_ = 1) was diluted to obtain a final concentration of 2 × 10^5^ cells in 100 µl per well of microtiter plates of 96-wells. Different concentrations of serially-diluted antibiotics were added to the wells in triplicates and the plate was incubated at 37 °C during 16 h. A negative control with plain Muller-Hinton broth medium was included in all experiments.

### 2.4. Macrophage Survival Test in the Presence of Different Antibacterial Compounds

Host cell infection assays were performed as described previously [22] using low-passage J774.A1 murine macrophages (ATCC) cultured in Dulbecco’s Modified Eagle Media (DMEM -Thermo-Fischer Scientific, Waltham, MA, USA). Briefly, macrophages were infected at a multiplicity of infection of 10 (MOI = 10) with *R. equi* 103S^+^. After 1 h of incubation, the medium was replaced with DMEM supplemented with 100 µg/mL gentamicin to kill extracellular bacteria. After 1 h of incubation in DMEM with gentamicin, cells were washed three times with PBS and incubated with DMEM supplemented with specific antibacterial compounds at different concentrations. After 8 h, macrophages were lysed with 0.1% Triton X-100 and serial dilutions of the lysates were spread onto TSA plates for *R. equi*’s colony forming unit (CFU) counting. The presence of the virulence plasmid pVAPA in *R. equi* was routinely verified by PCR amplification as previously described [17].

### 2.5. Statistical Analyses

Statistical analyses were conducted using IBM^®^ SPSS^®^ statistics v24 (IBM, ArmonK, New York, NY, USA). Firstly, data distribution was assessed with the Kolmogorov-Smirnov test. If data followed a normal distribution, two-way ANOVA and post-hoc Tukey’s multiple-comparison tests were employed to identify statistically significant differences across conditions. When the data were not following a normal distribution, a non-parametric analysis was performed employing a Kruskall-Wallis test.

## 3. Results and Discussion 

### 3.1. Identification of ROS-Generating Antibacterial Compounds Active against R. equi

*R. equi* is well equipped with antioxidant molecules and enzymes [22], which might confer a natural resistance to ROS-generating antimicrobials. Low molecular weight thiols are part of one of the most important antioxidant strategies in bacteria. Mycothiol (MSH) is a low molecular weight thiol unique of actinobacteria [23]. Mycothiol protects proteins from irreversible cysteine oxidation during oxidative stress by forming disulfides with protein thiols. Mycoredoxins (Mrxs) are then required for the reduction of S-mycothiolated proteins to restore their function. Mycoredoxins become S-mycothiolated during the process of de-mycothiolation, and mycothiol restores their reduced state by generating mycothione (MSSM) in the process. Mycothione is then reduced back to mycothiol by an NADPH-dependent mycothione reductase (Mtr).

The *mrx*-null mutant strain *R. equi Δmrx1Δmrx2Δmrx3* lacks three genes encoding for mycoredoxins, which made it highly susceptible to oxidative stress [17]. Due to this, we considered *R. equi Δmrx1Δmrx2Δmrx3* a very suitable reporter for screenings aimed at identifying ROS-generating antimicrobials. Thus, we compared the susceptibility of *R. equi* 103S^+^ and *R. equi Δmrx1Δmrx2Δmrx3* to different antibacterial compounds. 

We first analysed by the Kirby-Bauer disk diffusion method [24] the growth inhibition area generated by different antibiotics (Figure 1A). We did not observe any halos of inhibition when oxacillin, ampicillin, or fosfomycin trometamol were used in our screen, suggesting that the natural resistance of *R. equi* to these compounds was unrelated to mechanisms implicated in redox homeostasis. On the other hand, the susceptibility of *R. equi* 103S^+^ and *R. equi Δmrx1Δmrx2Δmrx3* to chloramphenicol and clindamycin was apparently analogous (Figure 1A), suggesting that these antibiotics did not generate ROS. However, the halo of inhibition generated by rifampicin, erythromycin, and vancomycin was significantly higher in the triple *mrx*-deletion mutant strain when compared to the wild type strain (Figure 1A). Interestingly, the combination of rifampicin and erythromycin is one of the most commonly used combinatorial antibiotherapy to treat *R. equi* infections [10]. Therefore, our results suggested that ROS synthesis might have an important role in the mechanism of action of this therapeutic strategy.

To confirm that our initial screening results were effectively due to ROS biosynthesis, we employed the well-characterized Mrx1-roGFP2 biosensor [17] to evaluate the oxidative stress generated in *R. equi* by very low doses of rifampicin, erythromycin, and vancomycin (Figure 1B). It has been recently demonstrated that the redox state of Mrx1-roGFP2 is in direct equilibrium with mycothiol/mycothione levels [17,19]. Moreover, Mrx1-roGFP2 could be used to evaluate in real-time the intracellular redox status of bacteria expressing the reporter [17].

Importantly, all three antibiotics generated clear oxidation peaks in Mrx1-roGFP2 (Figure 1B). In particular, we observed an oxidation peak starting at 5 min of exposure to rifampicin and erythromycin, whereas the oxidation peak caused by vancomycin was only detected after 10 min. These results suggested that the mechanism(s) of ROS biosynthesis of vancomycin might be different to those of rifampicin and erythromycin. In addition, the ratiometric responses of Mrx1-roGFP2 to these antibiotics validated our screening approach based on the high susceptibility of the *R. equi Δmrx1Δmrx2Δmrx3* strain to oxidative stress.

On the other hand, the redox potential (*E*_roGFP2_) of Mrx1-roGFP2 was −254 mV in response to rifampicin, −261 mV in erythromycin, and –273 mV in vancomycin. To put this in context, the basal redox potential of this biosensor was established recently as −290 mV [17], whereas the *E*_roGFP2_ of Mrx1-roGFP2 in response to 5 mM H_2_O_2_ was −264 mV and the intracellular redox potential during host cell infection was −260 mV. Overall, these results confirmed that rifampicin, erythromycin, and vancomycin elicited a significant oxidative stress as part of their mechanism of action. 

In an attempt to identify other ROS-generating antimicrobials with repurposing potential, we expanded the list of antibiotics tested in our screen with four additional compounds whose ROS-generating mechanism of action has been previously described in other pathogens: Artemisinin, clofazimine, norfloxacin, and nitrofurantoin [14,16,25] (Figure 2). We first compared the effect of these ROS-generating antimicrobials against *R. equi* 103S^+^ and *R. equi Δmrx1Δmrx2Δmrx3* by means of the disk diffusion method. We did not observe any halos of inhibition with artemisinin and nitrofurantoin, suggesting that the resistance of *R. equi* to these compounds is unrelated to redox stress. Interestingly, *R. equi* was susceptible to the ROS-generating anti-tuberculosis drug clofazimine [26] (Figure 2A). However, we did not observe any difference in the susceptibility of *R. equi Δmrx1Δmrx2Δmrx3* to this compound when compared to the wild type strain, which suggested that the mechanism of action of clofazimine in *R. equi* was not mediated by redox stress. Importantly, the zone of inhibition of norfloxacin was significantly higher for *R. equi Δmrx1Δmrx2Δmrx3* when compared to *R. equi* 103S^+^ (Figure 2A). Moreover, we observed a clear peak of oxidation in Mrx1-roGFP2 after only 3 min of exposure to norfloxacin (Figure 2B), and the redox potential of Mrx1-roGFP2 was −278 mV. Taken together, these results suggested that one of the mechanisms of action of norfloxacin was mediated by redox stress in *R. equi*, most likely by the generation of oxidized nucleotides within the bacterial cell [25,27,28].

### 3.2. Combinatorial Antibiotherapy against R. equi with ROS-Generating Antimicrobials

We considered our screening results as proof-of-principle of the feasibility of an antibiotherapy against *R. equi* based on antibiotics that generate oxidative stress. Consequently, we studied a possible synergistic effect of ROS-generating anti-*R. equi* compounds. We therefore analysed the combined effect of the most promising ROS-producing compounds against *R. equi* 103S^+^ in comparison to the commonly used combination of erythromycin and rifampicin (Figure 3). Interestingly, norfloxacin combined with either rifampicin, erythromycin, or vancomycin had an equivalent antimicrobial effect to the antibiotherapy based on erythromycin and rifampicin (Figure 3). This suggested that norfloxacin could be an important ROS-generating adjuvant to other anti-infectives against *R. equi*. In contrast, the oxidative stressing anti-tuberculosis drug clofazimine did not complement the antimicrobial effect of neither rifampicin nor erythromycin (Figure 3), confirming our screening results.

In order to produce a clinically relevant analysis of the combined effect of norfloxacin with any of the other ROS-generating antimicrobials tested, we calculated the minimum inhibitory concentration (MIC) of different combinations of antibiotics following the guidelines of the Clinical & Laboratory Standards Institute [10,21,29] (Table 1). 

We first established the MIC of individual antibacterial compounds (Table 1). Surprisingly, norfloxacin alone did not generate a reportable MIC for *R. equi* 103S^+^ below 10 µg/mL, despite being a broad-spectrum antimicrobial [30]. In addition, the combination of rifampicin and erythromycin did not generate a synergistic effect (Table 1). However, low doses of norfloxacin (1 µg/mL) combined with subminimal inhibitory concentrations of erythromycin (2.6-fold lower than its individual MIC) or vancomycin (2-fold lower than its individual MIC) inhibited *R. equi* growth (Table 1), which suggested synergy. Moreover, a similar effect was detected with combinations of low doses of vancomycin and either rifampicin (8-fold lower than its individual MIC) or erythromycin (2.6-fold lower than its individual MIC; Table 1). Overall, our in vitro results suggested that norfloxacin and vancomycin might be very promising ROS-generating agents against *R. equi*. However, vancomycin is considered a last-resort drug and therefore its veterinary use is not recommended.

### 3.3. Antimicrobial Activity of ROS-Generating Anti-Infectives against Intracellular R. equi

Antibacterial compounds might have deleterious effects on mammalian cells [15]. Therefore, the use of ROS-generating antimicrobials against intracellular pathogens might lead to unexpected results during host-cell infection. Due to this, we studied the combinatorial antimicrobial activity of the ROS-generating drugs identified in our screening on the intracellular survival of *R. equi* (Figure 4). In these experiments, we initially used the minimal inhibitory concentrations determined before for each antibiotic or combinations of antibiotics (Table 1). 

However, the synergistic effect of norfloxacin in combination with other ROS-producing antimicrobials such as erythromycin or vancomycin was only reproduced against intracellular *R. equi* when the concentration of this antibiotic was raised 20-fold (Figure 4). Similar to vancomycin [31,32], the cellular uptake of norfloxacin in macrophages might be quite poor, which could be an important barrier for the use of this therapeutic strategy in vivo. However, the intracellular delivery of antibiotics could be facilitated by their liposomal encapsulation [31]. Moreover, the therapeutic index of the encapsulated antibiotics increases, while the drug toxicity is reduced due to their intracellular delivery. Consequently, lower plasma concentrations are required to achieve an effective antimicrobial activity against intracellular pathogens [31]. Accordingly, gentamicin encapsulated in liposomes efficiently eradicated *R. equi* in a mouse model of infection [32]. Unfortunately, liposomal gentamicin caused nephrotoxicity in foals [33], abolishing the potential veterinary use of this particular encapsulated antibiotic. Nevertheless, the use of liposomal antibiotics for the treatment of infections caused by intracellular pathogens is now widely recognized as an important and safe therapeutic strategy to increase the internalization of highly effective antibiotics with poor cellular uptake [34]. 

## 4. Conclusions

*R. equi* is an intracellular veterinary pathogen that is rapidly acquiring antimicrobial resistance to currently used anti-infective strategies [7,8]. Rifampicin resistance in *R. equi* is associated to mutations of the *rpoB* gene [10], while the lateral acquisition of *erm* (46) gene is driving the emergence of macrolide-resistant *R. equi* isolates in the United States [35]. 

With the aim to develop novel ROS-inspired antimicrobial therapies against this actinobacterial pathogen, here we developed a new screening strategy for the *in vitro* identification of antimicrobial compounds that might synergistically cause oxidative stress. Interestingly, norfloxacin might be a promising adjuvant to other ROS-generating antimicrobials currently used to treat *R. equi* infections. Norfloxacin is a drug with great repurposing potential, as it is commonly used to treat urinary infections in humans [30]. Therefore, these results might pave the way for the rapid development of novel antimicrobial strategies with important veterinary applications.

## Figures and Tables

**Figure 1 antioxidants-09-00114-f001:**
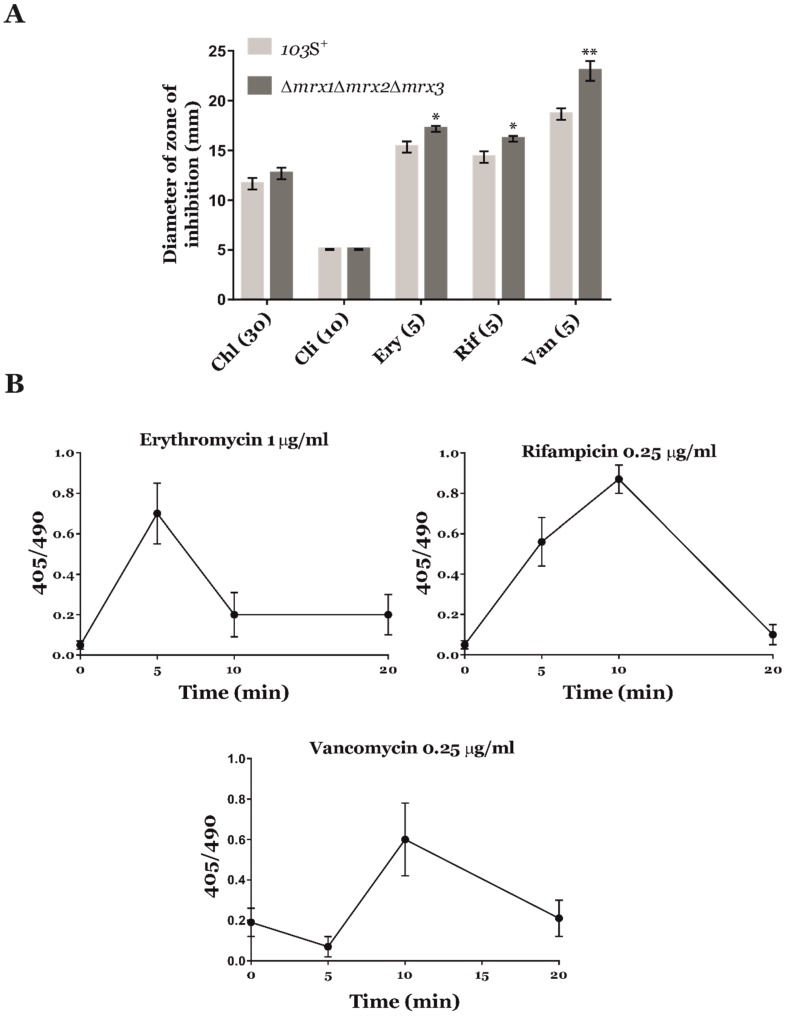
Screening results for the identification of ROS-generating antimicrobials active against *R. equi*. (**A**) Antimicrobial susceptibility of *R. equi* 103S^+^ (clear grey) and *R. equi Δmrx1Δmrx2Δmrx3* mutant (dark grey) to different antibacterial compounds: Chloramphenicol (Chl), clindamycin (Cli), erythromycin (Ery), rifampicin (Rif), and vancomycin (Van). The concentration used of each antibiotic is shown in brackets (µg/mL). The diameter of the growth inhibition zones was measured to the nearest millimetre, and the mean ± SD of three independent experiments was plotted. A Kruskall-Wallis non-parametric analysis was performed to test for statistical significance across comparisons of the wild type strain and the triple *mrx*-null mutant. *p*-value < 0.05 (*) or *p*-value < 0.01 (**). (**B**) Ratiometric response of the Mrx1-roGFP2 biosensor. Fluorescence 405/490 ratio was calculated by confocal microscopy at different time points in response to very low doses of rifampicin, erythromycin, and vancomycin. The results show the mean ± SD of three independent experiments.

**Figure 2 antioxidants-09-00114-f002:**
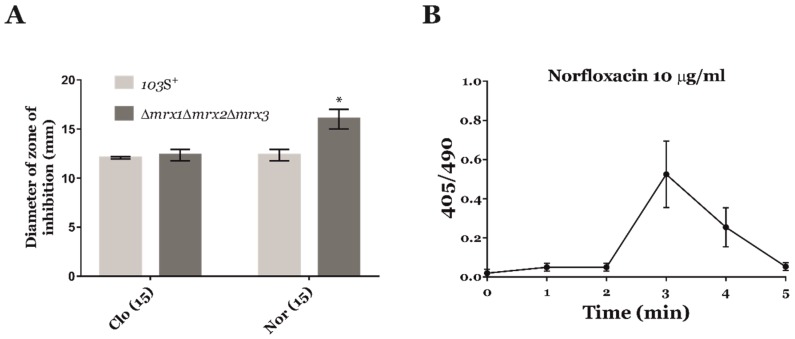
Identification of other ROS-generating antimicrobials against *R. equi*. (**A**) Antimicrobial susceptibility of *R. equi* 103S^+^ (clear grey) and *R. equi Δmrx1Δmrx2Δmrx3* mutant (dark grey) to clofazimine (Clo) and norfloxacin (Nor). The concentration of each antibiotic used is shown in brackets (µg/ml). The diameter of the growth inhibition zones was measured to the nearest millimetre, and the mean ± SD of three independent experiments was plotted. Two-way ANOVA and post-hoc Tukey´s multiple comparison tests were performed to assess for statistical significance across comparisons of the wild type strain and the triple *mrx*-null mutant. *p*-value < 0.05 (*). (**B**) Ratiometric response of the Mrx1-roGFP2 biosensor. Fluorescence 405/490 ratio was calculated by confocal microscopy at different time points in presence of 10 µg/mL of norfloxacin. The results show the mean ± SD of three independent experiments.

**Figure 3 antioxidants-09-00114-f003:**
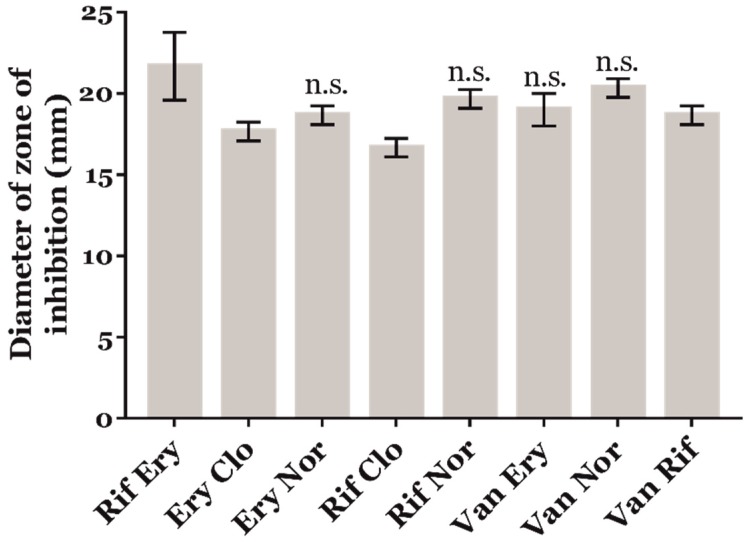
Antimicrobial susceptibility of *R. equi* 103S^+^ to 5 µg of different combinations of antimicrobial compounds, including clofazimine (Clo), erythromycin (Ery), norfloxacin (Nor), rifampicin (Rif), and vancomycin (Van). The diameter of the growth inhibition zones was measured to the nearest millimetre, and the mean ± SD of three independent experiments was plotted. Two-way ANOVA and post-hoc Tukey´s multiple comparison tests were performed to assess for statistical significance across conditions. N.s.: Not significant.

**Figure 4 antioxidants-09-00114-f004:**
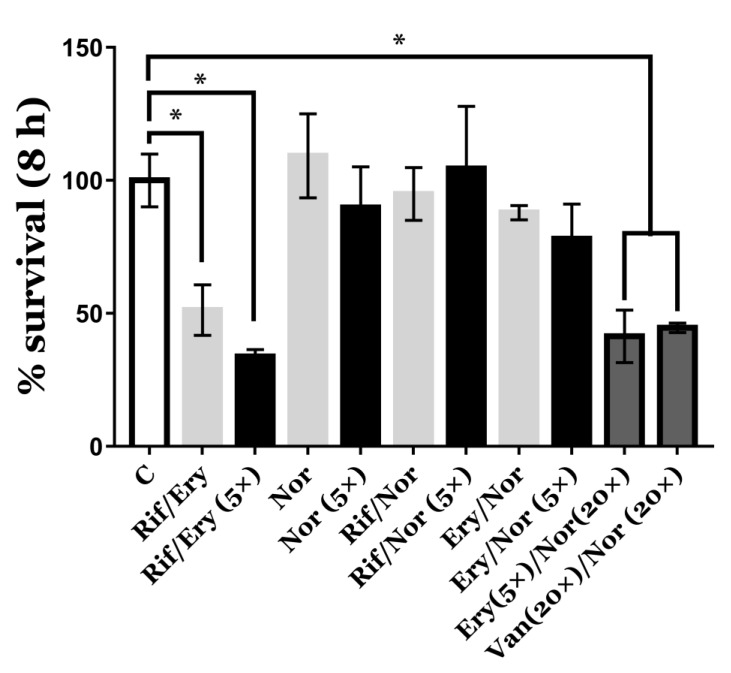
Antimicrobial effect of different combinatorial antibiotherapies against intracellular *R. equi* 103S^+^ infecting J774.A1 murine macrophages. Percentages of survival were quantified after 8 h of infection. Different combinations of rifampicin (Rif), erythromycin (Ery), norfloxacin (Nor), and vancomycin (Van) were compared against a negative control with gentamicin (C). The concentration of some drug combinations was increased 5-fold (5×) or 20-fold (20×) their MIC. Two-way ANOVA and post-hoc Tukey’s multiple comparison tests were performed to assess for statistical significance related to the number of CFU recovered from the negative control (C). *p*-value < 0.05 (*).

**Table 1 antioxidants-09-00114-t001:** MICs of different antibacterial compounds and their combinations against *R. equi* 103S^+^. The data are resulting from three independent experiments with two technical replicates per assay.

Antibiotics	MIC
Rifampicin	0.4 µg/mL
Erythromycin	1.3 µg/mL
Vancomycin	1 µg/mL
Norfloxacin	>10 µg/mL
Rifampicin + Erythromycin	0.4 + 1.3 µg/mL
Rifampicin + Norfloxacin	0.4 + 1 µg/mL
Erythromycin + Norfloxacin	0.5 + 1 µg/mL
Rifampicin + Vancomycin	0.05 + 0.25 µg/mL
Erythromycin + Vancomycin	0.5 + 0.25 µg/mL
Vancomycin + Norfloxacin	0.5 + 1 µg/mL

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
