# Peer review of "A Novel Screening Strategy Reveals ROS-Generating Antimicrobials That Act Synergistically against the Intracellular Veterinary Pathogen Rhodococcus equi"

_antioxidants, 2020, doi:10.3390/antiox9020114_

Round 1

Reviewer 1 Report

The manuscript  ID: antioxidants-697460 presents interesting results of studies on the synergistic efficacy of selected ROS-generating antimicrobials against Rhodococcus equi. An increasing antimicrobial resistance is observed  in this animal pathogen therefore each findings that may improve therapeutic options used for R. equi infection treatment are important and desired. However the presented results should be considered as preliminary because these in vitro studies were performed only for one wild strain (R. equi 103S+) in comparison to the mutant strain.

The manuscript is well written, however I suggest some changes in this work.

Major comments:

1) I propose to reword the title – it indicates that new therapeutic options, well documented, including the in vivo studies, are described in the work, while the presented results rather refer to the development of a new screening strategy for the in vitro identification of ROS-generating antimicrobials that may act synergistically.

2) The studies were focused on an antimicrobial activity associated with ROS-generating and in this context susceptibility/resistance to selected antimicrobials was determined. However I think that information about other mechanisms of resistance to tested antimicrobials occurring in both studied strains, R. equi 103S+ and its derivative mrx1-3 mutant, should be included in the manuscript.

3) The results of antimicrobial susceptibility testing performed by disk diffusion method (Figures 1A and 2A) should be confirmed by MIC assay, as a broth/agar dilution method is recommended for R. equi.

4) Why the in vitro antibacterial activity of various combinations of antimicrobials was performed by disk diffusion method and MIC assay only for the wild type strain? It could be interesting to compare the results for both studied strains.

5) Lines 215-216: This conclusion should be corrected because it was not well documented – only one wild strain was tested (moreover it is susceptible to most of tested antimicrobials).

Regarding vancomycin as a promising antimicrobials against R. equi - it should be considered that this antimicrobial belongs to last resort drug and its use in veterinary should  not be recommended.

Minor comments:

1) Information about R. equi mutant strain should be added to the 2.1 section.

2) Disk diffusion and MIC interpretative criteria (breakpoints) for tested antimicrobials, used in the study should be included into Materials and Methods section.

3) Lines 97 – 135 – please use italic letters in case of bacterial names.

4) Lines 112- 113 – the results for oxacillin, ampicilin and fosfomycin are not showed in Figure 1A.

5) Line 168: should be - ...these results suggest that one of the mechanisms of action...

6)Line 172 (Fig. 2): It would be better “other ROS-generating” then “novel ROS-generating”.

7)Line 225 (Fig. 4): R. equi strain designation should be added.

8) Small linguistic corrections in the manuscript  are needed.

Author Response

Many thanks to your comments, the manuscript has greatly improved after we addressed all of your points. Please find below our answers.

The manuscript  ID: antioxidants-697460 presents interesting results of studies on the synergistic efficacy of selected ROS-generating antimicrobials against Rhodococcus equi. An increasing antimicrobial resistance is observed  in this animal pathogen therefore each findings that may improve therapeutic options used for R. equi infection treatment are important and desired. However the presented results should be considered as preliminary because these in vitro studies were performed only for one wild strain (R. equi 103S+) in comparison to the mutant strain.

The manuscript is well written, however I suggest some changes in this work.

Major comments:

1) I propose to reword the title – it indicates that new therapeutic options, well documented, including the in vivo studies, are described in the work, while the presented results rather refer to the development of a new screening strategy for the in vitro identification of ROS-generating antimicrobials that may act synergistically.

We agree with the reviewer, the title has been amended following this comment (Lines 2-3).

2) The studies were focused on an antimicrobial activity associated with ROS-generating and in this context susceptibility/resistance to selected antimicrobials was determined. However I think that information about other mechanisms of resistance to tested antimicrobials occurring in both studied strains, R. equi 103S+ and its derivative mrx1-3 mutant, should be included in the manuscript.

We have added some information about the molecular basis of resistance to rifampicin and macrolides in the conclusions (Lines 253-255). However, very little is known about the specific mechanisms of resistance in R. equi to the remaining drugs tested in this manuscript.

3) The results of antimicrobial susceptibility testing performed by disk diffusion method (Figures 1A and 2A) should be confirmed by MIC assay, as a broth/agar dilution method is recommended for R. equi.

Many thanks for this comment, but we could not do any further experiments due to the very short period of time given to do this revision. However, the results obtained with the disk diffusion method allowed us to identify the antibiotic drugs that act synergistically in the MIC assays reported in Table 1, which could be considered an indirect validation of the results presented in Figures 1A and 2A.

4) Why the in vitro antibacterial activity of various combinations of antimicrobials was performed by disk diffusion method and MIC assay only for the wild type strain? It could be interesting to compare the results for both studied strains.

Yes, the susceptibility pattern of R. equi Δmrx1Δmrx2Δmrx3 could be informative to some extent. However, our main focus was to present data with clinical relevance, and because of this reason these experiments were only done with the wild type strain.

5) Lines 215-216: This conclusion should be corrected because it was not well documented – only one wild strain was tested (moreover it is susceptible to most of tested antimicrobials).

Regarding vancomycin as a promising antimicrobials against R. equi - it should be considered that this antimicrobial belongs to last resort drug and its use in veterinary should  not be recommended.

Many thanks for raising these two points, we have amended that conclusion following your comment (Line 219), and we have mentioned the limitation of the use of vancomycin in veterinary practice in lines 219-220.

Minor comments:

1) Information about R. equi mutant strain should be added to the 2.1 section.

The requested information is now included in lines 52-53.

2) Disk diffusion and MIC interpretative criteria (breakpoints) for tested antimicrobials, used in the study should be included into Materials and Methods section.

Unfortunately, interpretive criteria for the classification of isolates of R. equi as susceptible, intermediate, or resistant are still not available (please see Microbiol Spectrum 5(5):ARBA-0004-2016 and http://www.eucast.org/clinical_breakpoints/).

3) Lines 97 – 135 – please use italic letters in case of bacterial names.

Our apologies, this is now corrected.

4) Lines 112- 113 – the results for oxacillin, ampicilin and fosfomycin are not showed in Figure 1A.

We did not observe any reportable inhibition zones with these antibiotics.

5) Line 168: should be - ...these results suggest that one of the mechanisms of action...

Many thanks, this is now corrected (line 169).

6)Line 172 (Fig. 2): It would be better “other ROS-generating” then “novel ROS-generating”.

We have modified this sentence following the reviewer’s suggestion (Line 173).

7)Line 225 (Fig. 4): R. equi strain designation should be added.

Many thanks, this is now corrected (Line 230).

8) Small linguistic corrections in the manuscript  are needed.

We have amended many small linguistic mistakes; all of these changes are now tracked in the new version of the manuscript.

Reviewer 2 Report

"This work reports a new therapeutic approach against the intracellular veterinaty pathogen Rhodococcus equi,based on the combination of macrolides with ROS-generating antimicrobial compounds. The suggested novel ROS-inspired antimicrobial therapy has been developed by an efficient new screening strategy for the in vitro identification of antimicrobial compounds active against R. equi. The possible synergistic effect of ROS-generating anti-R. equi compounds has been investigated by analyzing the combined effect of the most promising ROS-producing compounds, in comparison to the commonly used combination of erythromycin and rifampicin. In this context, norfloxacin has been identified as a promising adjuvant to other antimicrobials compounds, that may synergistically cause oxidative stress, currently used used to treat R. equi infections. The paper is well developed and the obtained results appear promising. At my opinion, the paper can be accepted for publication."

Author Response

Many thanks for your kind comments.